# Discharge communication study: a realist evaluation of discharge communication experiences of patients, general practitioners and hospital practitioners, alongside a corresponding discharge letter sample

Katharine Weetman ![ORCID],[1] Jeremy Dale ![ORCID],[1] Emma Scott,[1] Stephanie Schnurr[2]

[1]Unit of Academic Primary Care, Warwick Medical School, University of Warwick, Coventry, UK
[2]Applied Linguistics, University of Warwick, Coventry, UK

**Correspondence to**
Dr Katharine Weetman;
Katharine.Weetman@warwick.ac.uk

## ABSTRACT

**Objectives** To develop a programme theory for the intervention of patients receiving discharge letters.

**Design** We used a realist evaluation approach and captured multiple perspectives of hospital discharge to refine our previously developed programme theory. General practitioner (GP), patient and hospital clinician views of a single discharge event in which they were all involved were collected using semi-structured interviews and surveys. These were then triangulated to match the corresponding discharge letter. Data were qualitatively synthesised and compared in meta-matrices before interrogation with realist logic of analysis to develop the programme theory that maps out how patients receiving discharge letters works in specific contexts.

**Setting** 14 GP practices and four hospital trusts in West Midlands, UK.

**Participants** 10 complete matched cases (GP, patient and hospital practitioner), and a further 26 cases in which a letter was matched with two out of the three participants.

**Results** We identified seven context mechanism outcome configurations not found through literature searching. These related to the broad concepts of: patient preference for receiving letters, patient comprehension of letters, patient-directed letters, patient harm and clinician views on patients receiving letters. 'Patient choice' was important to the success (or not) of the intervention. Other important contexts for positive effects included: letters written in plain English, lay explanations for jargon, verbal information also provided, no new information in letter and patient choice acknowledged. Three key findings were: patient understanding is perhaps greater than clinicians perceive, clinician attitudes are a barrier to patients receiving letters and that, negative outcomes more commonly manifested when patients had not received letters, rather than when they had.

**Conclusions** We suggest how patients receiving discharge letters could be improved to enhance patient outcomes. Our programme theory has potential for use in different healthcare contexts and as a framework for policy development relating to patient discharge.

## Strengths and limitations of this study

► First study to compare and contrast matched views of patients, general practitioners (GPs) and hospital clinicians in relation to specific discharge letters.
► Realist theory facilitated understanding of not just whether patients should receive letters, but how this practice may 'work' in different contexts and why.
► The qualitative methods enabled detailed gathering of the experiences, viewpoints, and attitudes of participants.
► The secondary analysis was limited by weaknesses in the primary dataset, including the sociodemographic diversity of the patients, range of conditions, and the limited numbers of cases in which hospital clinician perspectives could be matched to those of GPs and patients.
► Evidence relating to children, mental health admissions, and those lacking capacity was not considered.

## INTRODUCTION

### Background

Effective communication during discharge care transitions is essential for patient safety and to reduce negative outcomes[1] such as hospital readmissions.[2] Despite this, studies[3–5] continue to reiterate that processes and content of discharge communication require improvement. Internationally, the practice of patients receiving letters varies but it is generally common for hospital doctors to write directly to general practitioners (GPs) or equivalent.[6] UK standards and policies[7–11] currently outline that patients should receive copies of letters between physicians as a 'right'[11] and that this is 'good practice',[7] unless there is a risk of harm. Initiatives such as 'please write to me'[8] by the *Academy of Medical Royal Colleges* have sought to increase

practice of patients receiving letters and suggested modifications such as using plain English to increase patient comprehensibility. A recent review by Rayner *et al*[6] highlighted the value of writing to patients in order to enhance collaborative working and positive outcomes. Despite this, research,[12–14] both within the UK and internationally, continues to report that patients receive letters inconsistently, the effects of which are unclear.[14 15] Reasons for this inconsistency are little understood but physician attitudes such as concerns about perceived harm may be acting as a barrier to policy uptake which has implications for patient experience and safety.[14] It is important to understand the extent to which this occurs purposefully, and how this affects patient experience and outcomes.

Our previous realist review[14] found conflicts between clinician and patient perspectives in relation to patients receiving discharge letters (eg, perceived rates of patient understanding). Hence, the current study was designed to shed light on reasons for conflicts through investigating experiences from multiple viewpoints within the same discharge events. The objectives were to undertake an investigation of how patients receiving discharge letters may be improved alongside best practice recommendations and to develop a programme theory for patients receiving letters. As outlined in the works of Pawson,[16–19] a 'programme theory' is useful as it goes beyond consideration of 'does it work' and instead seeks to explain *how* an intervention may be theorised to 'work' to include within what contexts, for whom, why and to what extent.[16 20] The research questions were:

1. How do the experiences of patients, GPs and hospital practitioners (HPs) differ and align within the multi-perspective discharge communication cases?
2. How does patients receiving discharge letters work (or not) and what are the important contexts associated with the desired positive effects?

This is the final paper in a series forming the Discharge Communication Study[21]; the others are summarised in box 1. Results relating to the GPs and patients are published.[22 23]

## METHODS
### Study design
This study was a secondary analysis of a subset of data from the Discharge Communication Study, an exploratory mixed methods study based in the West Midlands, UK[21]; box 1 gives a brief summary of papers linked to the Discharge Communication Study. The intervention under scrutiny 'patients receiving discharge letters' was defined by the team as 'the patient being given or sent any form of written (paper or digital) hospital discharge communication; this could be a direct copy, patient-directed letter, or a combination'. Broadly, the data comprised three elements: (i) GP sampling and rating of discharge letters ('successful' or 'unsuccessful') and narrative interviews, (ii) semi-structured interviews with patients to whom the letters related, (iii) survey of HPs who wrote the sampled letters.

---

**Box 1  Summary of discharge communication studies and results**

**GP study**[22]

**Methods**
► 53 GPs were recruited from 18 practices within the West Midlands (UK) through the local Clinical Research Network and Warwick Medical School links with practices.
► They were asked to purposively sample[24] 14–24 recent (<3 weeks) discharge letters in accordance with the inclusion and exclusion criteria (see table 1).
► Each GP completed a discharge letter selection template (see online supplemental file 1) with their discharge letter grading (successful or unsuccessful) and their comments.
► A subgroup of 26 GPs took part in an audio-recorded interview or focus group; these took place face to face at GP practices and over the telephone (see online supplemental file 2 for interview guide).

**Main findings**
► Key components within discharge letters (eg, GP actions) associated with successful gradings.
► The importance of clarity and comprehensibility.

**Patient study**[23]

**Methods**
► The patients associated with each of the letters sampled by GPs were invited to take part in a 1-1 semi-structured interview at their home or GP surgery (see online supplemental file 3 for interview guide).
► No relationship was established with participants prior to the study.
► All interview/focus group data were audio-recorded and transcribed by KW who also took notes. Transcripts were not shown to participants.

**Main findings**
► 50 patients to whom the sample letters related took part in interviews.
► They generally wanted to receive copies of their discharge communication letter.
► Patients also suggested how letter comprehensibility may be improved (eg, no acronyms).

**Hospital practitioner study**

**Methods**
► The hospital practitioners who wrote the letters sampled by GPs were invited to take part in a survey.

**Main findings**
► 46 hospital practitioners completed surveys.
► There were differences between what clinicians felt should be done and what occurred in practice, for example, 26 (56.5%) felt patients should always receive letters and 17 (37.0%) did this in practice.
► Some hospital practitioners expressed reservations around patients receiving letters.
► Many were unaware of the Department of Health guidelines on copying letters to patients.[7]

GP, general practitioner

---

### Settings
The setting for the study is outlined in the published study protocol.[21] It involved four hospital trusts and a diverse range of 18 GP practices in the West Midlands.

| Table 1 | Discharge letter inclusion and exclusion criteria |
|---|---|
| Inclusion criteria | ▶ NHS adult (18+ years) patients recently discharged (≤3 weeks) from hospital following an episode of inpatient or outpatient care. <br> ▶ Patient registered with the participating GP practice. <br> ▶ Patient treated at and discharged from hospital trusts within Warwickshire, Coventry, Rugby, Herefordshire, and Worcestershire. <br> ▶ Cases where written discharge communication has been sent to the patient's GP. |
| Exclusion criteria | ▶ Age <18 years. <br> ▶ Patients who lack capacity to give informed consent to participate in the study (eg, Alzheimer's, severe mental illness, etc) or are deemed by the GP to be unsuitable for participation (eg, end of life). <br> ▶ Patients discharged to providers or units other than their GP (eg, discharge from hospital to a rehab unit). <br> ▶ Discharge communication from mental health services. <br> ▶ Communication about individuals who are considered unable to participate in an interview or focus group or survey conducted in English. <br> ▶ Letter relates to patient who has expressed a general wish not to participate in research. |

GP, general practitioner; NHS, National Health Service.

### Recruitment and data collection

Recruitment and data collection took place, as detailed in previous publications[21–24] between August 2017 and September 2018. In brief, GPs were asked to screen (see table 1 for screening criteria) and select a sample of recently received discharge letters according to what they considered to be 'successful' or 'unsuccessful' letter exemplars; for each letter, GPs were asked to complete the selection proforma (online supplemental file 1) and rate the letters 'successful' or 'unsuccessful'.[24] There were no set criteria for letter ratings as the selection was based on each participating GP's interpretation of what makes a successful or unsuccessful discharge letter.[24] This purposive[25] letter sampling approach was intended to increase sample diversity and address the research questions within dichotomous contexts. All GPs involved in letter sampling were then invited to take part in a 'narrative'[26] interview or focus group with KW (see online supplemental file 2 for guide). All patients associated with the sampled discharge letters were sent an invitation pack by their GP practice; this invited them to take part in an audio-recorded semi-structured interview with KW (see online supplemental file 3 for interview guide). Finally, the hospital professionals who wrote or signed the sampled discharge letters were sent an invitation pack by the research team; this invited them to take part in a survey on their evaluation of the discharge letter they wrote, their current practices and their views about how discharge communication processes may be improved.[24] Packs were sent by post and email as well as being internally distributed by hospital sites.

For this study, we reinterpreted data collected across all of the other studies. This involved a secondary analysis of a subset of the data which was drawn from sampled discharge letters that could be 'matched' to at least two other dataset perspectives. Study specific ID codes allocated to the letters allowed cross-matching with participants to build multiple viewpoint cases termed 'quartets' (mapping together four elements if complete, or 'trios' if only one perspective missing—see figure 1).

The target was to build 30 quartet cases through recruiting at least 30 GPs, patients and HPs (target n=90). Trio and quartet participants were not separately recruited from other studies within the project; instead, cases were built through the participant recruitment and data collection across all studies for the discharge communication project (see figure 2). Once participant data across studies were matched into trio and quartet cases, findings and data were subjected to a secondary level data analysis using a realist approach described below. This allowed highlighting of data convergence and divergence as well as the emergence of new findings which only became apparent through juxtaposition.

### Analysis

The study was underpinned by a critical realist framework[27] and a generative view of causation, that is, not just whether an intervention works but in what contexts, how, for whom and why.[20] A realist logic of analysis[16–18 27] has the potential to account for complexity; discharge communication is complex in many ways such that the letter form (ie, typed or handwritten) and format (ie, narrative or

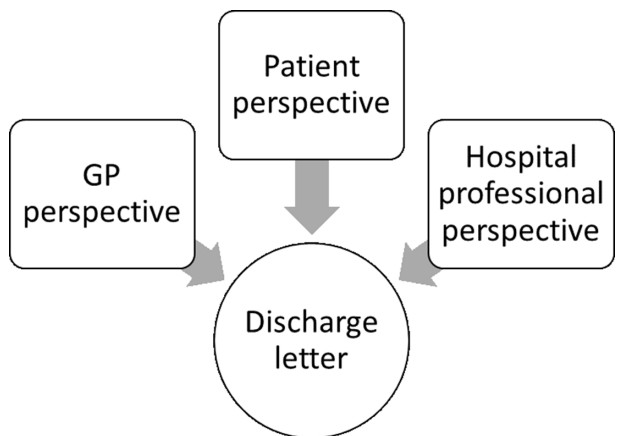

**Figure 1** Multiple-perspective 'quartet' case wherein comparisons occur between experiences associated with the same discharge letter. GP, general practitioner.

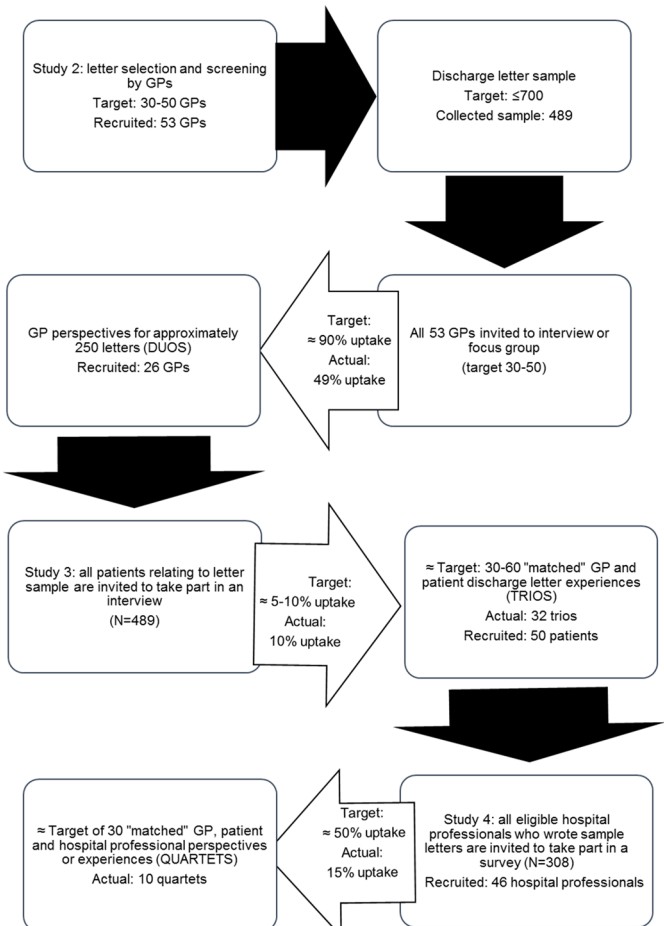

**Figure 2** Recruitment uptake across studies for the project to show how trio and quartet cases were formed. GP, general practitioner.

templated) as well as the communicative abilities and attitudes of both writers and recipients may vary. This study took a pragmatic approach to realist evaluation[17 28 29] in order to apply realist logic to multiple perspective cases within single discharge events. The study drew on realist principles to generate a 'programme theory' or theorised explanation of whether or not patients receiving letters 'works' (or not) as well as outlining the important relating context (C), mechanism (M) and outcome (O) configurations (CMOCs). The programme theory from our previously conducted realist review[14] was used as the starting theory; this was further developed based on the primary data results and findings. Interrogation and synthesis of evidence for CMOCs used a realist analytic approach[18] to consider the same theory of whether or not 'patients receiving letters' works in comparative settings.[30] Thus, analysis was grounded on the assumption that 'outcomes' of the intervention may vary according to 'context'.[30] All data were inspected for evidence of 'relevance'[20 30 31] to the theory. Manual note-taking on data was then undertaken[14] and judgements were formed as to what any new CMOCs might plausibly be prior to integration into the programme theory.

Data relating to each group was initially analysed separately (see box 1). Findings across groups were then triangulated and a secondary analysis was undertaken using meta-matrices to compare and contrast data. Such triangulation has previously been used within healthcare research,[32 33] particularly in relation to healthcare consultations,[34–36] to compare multiple perspectives. Multiperspective case analysis involved re-review of the data for each case; findings from different participants within letter cases were re-read and juxtaposed to highlight agreements and disagreements. Narrative summaries for each case were then developed. Summaries were not intended to be comprehensive but to select and include findings of relevance to the research questions. Analysis sought to reconcile previously identified literature disparities on this topic (see our realist review[14]) through highlighting source convergence and divergence in relation to 'patients receiving letters'.

### Patient and public involvement
Around 30 patients were involved in the research design through identifying research priorities[37] by 'ranking' potential research questions through completing surveys and taking part in discussions. Four persons with experience as carers from a pre-established panel also provided feedback on the readability and clarity of the patient information materials.

## RESULTS
### Recruitment
Figure 2 shows how data collection across all studies for the discharge communication project led to the formation of 26 trio cases (1 GP and HP, 3 patients and HPs, 22 patients and GPs) and 10 quartet cases (patient, GP and HP). Table 2 summarises the data characteristics in terms of GP grading, patient sex and age, discharge episode type (inpatient, outpatient …), specialty and HP grade. The 10 quartet cases had an even divide of GP graded successful and unsuccessful letters. Four patients reported that they had previously received the discharge letter and six reported that they had not. Letters related to six specialties across four hospital trusts.

### Context mechanism outcome configurations
Narrative summaries for our data are in online supplemental file 4 (trios) and online supplemental file 5 (quartets). Following a realist approach, findings were interrogated for theories and CMOCs of 'relevance'[20 30 31] to patients receiving discharge letters. The following section describes the identified CMOCs and concepts. Subheading themes which structured our realist review[14] were used and iteratively modified. The 48 CMOCs from the realist review were also systematically interrogated in light of the new evidence; 7 new CMOCs were added. The final table of 55 CMOCs is in online supplemental file 6.

**Table 2** Trio and quartet characteristics

| Characteristic | Trio cases (n=26) | Quartet cases (n=10) |
|---|---|---|
| GP grading | Successful: 18 (69.2%)<br>Unsuccessful: 8 (30.8%) | Successful: 5 (50.0%)<br>Unsuccessful: 5 (50.0%) |
| GP practices and GPs (n) | 14 GP practices, 17 GPs | 8 GP practices, 9 GPs |
| Practice sizes | Small (<5000 patients): 1 (7.1%)<br>Medium (5–10 000 patients): 8 (57.1%)<br>Large (10 000+ patients): 5 (35.7%) | Small (<5000 patients): 0 (0.0%)<br>Medium (5–10 000 patients): 4 (50.0%)<br>Large (10 000+ patients): 4 (50.0%) |
| Patient age | Range: 27–87<br>Median: 67 | Range: 59–77<br>Median: 71 |
| Patient sex | Female: 14 (53.8%)<br>Male: 12 (46.2%) | Female: 3 (30.0%)<br>Male: 7 (70.0%) |
| Admission | Inpatient: 20 (76.9%)<br>Outpatient: 2 (7.7%)<br>Other*: 4 (15.4%) | Inpatient: 7 (70.0%)<br>Outpatient: 1 (10.0%)<br>Other*: 2 (20.0%) |
| Specialties | 1. Urology: 2 (7.7%).<br>2. Respiratory: 1 (3.8%).<br>3. Accident and emergency: 4 (15.4%).<br>4. General surgery: 3 (11.5%).<br>5. Cardiology: 2 (7.7%).<br>6. Trauma and orthopaedics: 4 (15.4%).<br>7. Head and neck: 1 (3.8%).<br>8. Endocrinology: 1 (3.8%).<br>9. Plastic surgery: 1 (3.8%).<br>10. Neurosurgery: 1 (3.8%).<br>11. General medicine: 4 (15.4%).<br>12. Internal medicine: 1 (3.8%).<br>13. Renal medicine: 1 (3.8%). | 1. Urology: 3 (30.0%).<br>2. Respiratory: 2 (20.0%).<br>3. Accident and emergency: 1 (10.0%).<br>4. General surgery: 2 (20.0%).<br>5. Cardiology: 1 (10.0%).<br>6. Trauma and orthopaedics: 1 (10.0%). |
| Hospital grade of discharging physician | 2 grade types:<br>Consultant: 20 (76.9%)<br>Core trainee or equivalent: 6 (23.1%) | 4 grade types:<br>Consultant: 6 (60%)<br>Advanced clinical practitioner: 1 (10%)<br>Junior doctor: 2 (20%)<br>Senior house officer: 1 (10%) |

*Other may include but not be limited to admission types such as accident and emergency visit, day case procedure, or specialty assessment unit visit.
GP, general practitioner.

## Patient preference/choice

Of the 36 cases, 26 patients had received the discharge letter and 10 had not. Patients frequently emphasised positive effects of receiving letters such as increased satisfaction and a sense of involvement[12 38] (CMOC2). Patients explained that receiving letters can increase their autonomy and so encourage them to take control and 'ownership' of their health (CMOC5, CMOC14). In cases where patients had not received letters (C–E, H–J), patients reported difficulty retaining information and feeling unclear about what happened, their condition, and how to manage it. On the other hand, in cases where patients had received letters (context, C) (A, B, F, G), patients reported feeling informed and finding the letter useful as a reminder (mechanism, M) of what happened to increase recall[39 40] (outcome, O) (CMOC15) and decrease the need to memorise information (CMOC50).

Past studies, across a range of settings, report that patient preference for receiving letters is high (79%–97%)[39–46]; this study supports this finding as patients generally indicated preference for discharge letter receipt. Despite this, both GPs and patients noted the inconsistent practice of patients receiving letters. A potential suggested solution was for letters to contain a template 'tick box' (C) as to whether or not the patient has been given a letter copy so that it can be audited (O) and increase awareness of the practice (M) (CMOC49). One new CMOC that emerged was that patients may use the letter (M) as a record (C) for providing evidence for administrative proceedings (O) (eg, benefits) (CMOC51) or for care within unfamiliar settings (eg, holidays). Broadly, impacts on patients' experiences were framed as more positive when patients had received discharge letters and more negative when they had not. Crucially, positive outcomes were typically only

triggered within key contexts (eg, letter factually accurate (CMOC15)). Our realist review found patients generally did not object to social habits being included in the letter as long as it had relevance[14]; our findings here caveated this notion in that this information should also be phrased with neutral non-judgemental language (C) to reduce likelihood of upset (M) which could diminish wellbeing (O) (CMOC53). Crucially, patient preference was not 100% and it is important to consider those who may not wish to receive letters (CMOC40) through acknowledgements of *patient choice*[12 41–43] (CMOC41). Moreover, some patients may want to receive letters some of the time but not for every single care episode; patients identified this may apply in cases of repeat admissions for the same condition (C) where letters may be repetitive and not helpful (M) and so not requested (O) (CMOC52). Systems of letter receipt must therefore account for individual case variation.

## Patient comprehension

Findings supported previous evidence,[41 45 47 48] that patients may understand their letters (M) leading to improved patient knowledge and recall (O) as well as patients feeling empowered to take responsibility for their own health and so carrying out recommendations (CMOC12–CMOC15, CMOC54). However, letters are not always stylistically tailored to patients' needs due to the presence of medical jargon and acronyms. Within some cases (eg, case 6), the patient and GP agreed that the patient would have benefitted from use of lay terms in the letter to unravel the medical jargon. Case 5 highlighted that unexplained acronyms should be avoided for the sake of both patient and GP comprehensibility. There is a risk that patients receiving letters (C) may increase appointments or queries (O) as patients seek explanations of the letter contents (M).[49] Nevertheless, in line with past work,[46 50] findings were that this rarely occurs and indeed no study patients reported having made appointments for this purpose (CMOC7, CMOC11). Furthermore, patients reported that the absence rather than receipt of the letter is what would prompt them to visit the GP (M) and thus increased patient information (C) may reduce rather than increase appointments (O) (CMOC11). GPs suggested use of a 'patient information' section on the letter (C) which provides a letter synopsis in the form of a lay summary to increase understanding (M) and improve patient knowledge and satisfaction (O) (CMOC54). Patients and GPs agreed that letters should complement rather than substitute verbal information. This is seen in case 17 where the letter communicates a serious diagnosis to the patient and they report being given no other information from the hospital. Hence, letters should only be provided in the context of adequate patient counselling so that the letter is not communicating new information.

## Personalised or patient-directed discharge letters

Personalised letters may increase resource use and workload[45 48 51] (CMOC25). There were disagreements as to whether it would be more beneficial for patients to receive a separate personalised letter or the same letter as the GP; some clinicians felt personalised letters may improve patient comprehension (eg, case 1) whereas patients generally preferred to receive the same copy as the GP for transparency and reassurance (eg, cases 3, 22, 23) (CMOC26). Patients did suggest letter improvements in cases where the clinicians rated the letter successfully (cases B, I); patients felt letters should contain more information regarding how they can improve their condition and recommended patient actions.

## Patient harm

Clinicians sometimes had concerns that patients receiving letters may cause harm such as patient anxiety or confusion. However, clinician concern was expressed in several cases where the patients emphasised the benefits of discharge letter receipt (cases B, C, E, G, H). Patients suggested that receiving letters (C) may reduce negative outcomes through reassuring them and reducing or settling anxiety (M) thereby supporting their well-being (O) (CMOC39) (case 8). Instances which subverted this trend primarily related to the letter quality (eg, letter inaccuracies caused stress). One patient found that clear written information in bad news contexts (C) was particularly useful (M) as it allowed them to make an informed end of life plan (O). Suggestions to reduce risk of harm included ensuring the content is wholly factual and ensuring the patient consents to letter receipt[52] (CMOC41).

## Clinician views

Supporting past literature, some clinicians were in favour[50 53] (CMOC5, CMOC16) of the practice while others had reservations[12 47] (CMOC6, CMOC35). GPs appeared to be more in favour than HPs. Nonetheless, some GPs did express issues regarding the inherent need of letters to contain technical information which may not be patient comprehensible. HP concerns included: patient confusion and anxiety[13 38 44] (CMOC19), that the patient will not find the letter useful, that letters would need to be oversimplified,[12 54] and that receiving a letter may not be in the best interests of the patient (eg, mental health cases). Clinician and GP perceived benefits (CMOC5) of patients receiving letters were: increased sense of patient inclusion, improved understanding or knowledge,[51 54] and increased transparency[47] (CMOC33). Our realist review[14] suggested that patient understanding of their letters may be higher than clinicians perceive; this study further supports this notion. Comparably to previous literature, concern regarding 'patient understanding' was common[12 38 47 54] (CMOC6). However, clinician and patient views were sometimes the antithesis of one another; there were cases where the clinician had concerns (C) regarding patient comprehensibility (M) in cases where the patient reported to have found the letter useful (O) (CMOC55) (see cases A–C, E, G–H, J). Patients demonstrated resourcefulness through expounding that unknown terms can be looked up on

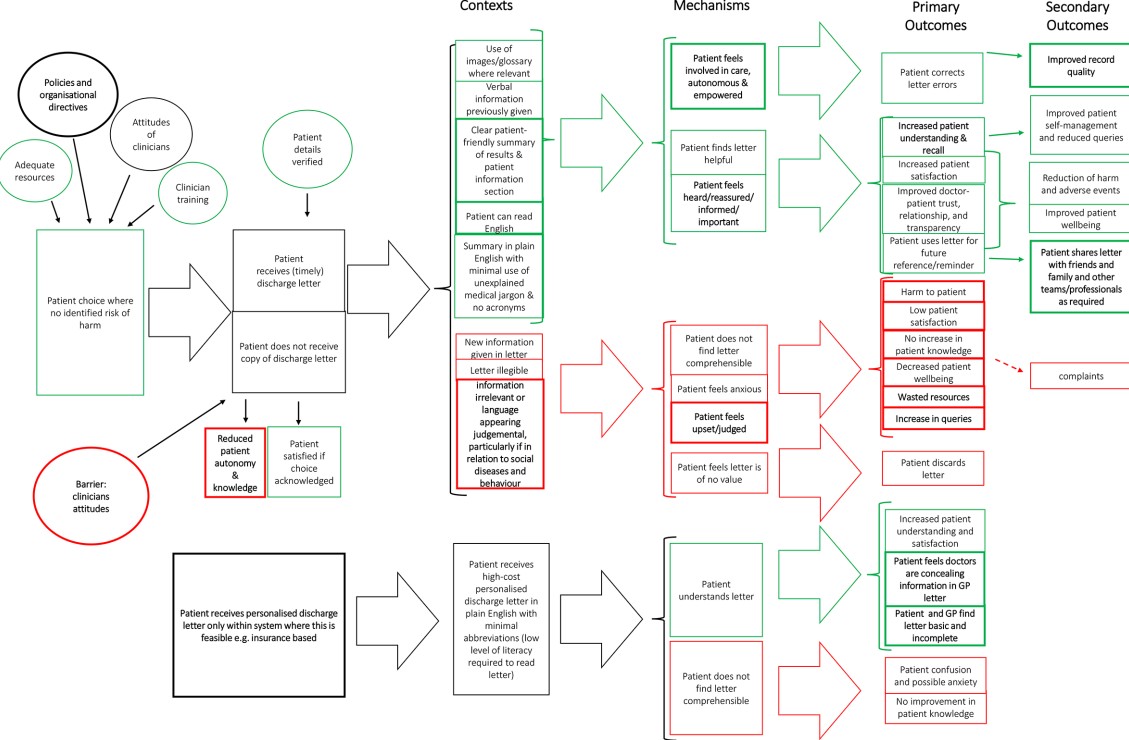

**Figure 3** Resultant programme theory that maps out how patients receiving discharge letters works (or not). GP, general practitioner.

the internet (case 19) as well as discretion (C) through appreciating that understanding the contents and implications (O) may not necessarily involve comprehending every word (M).

## Programme theory

Our findings were used to refine the programme theory, using our realist review[14] as the starting point; changes made to the theory are highlighted in bold (see figure 3). All matched cases were re-read, annotated, and interrogated for evidence. Relevant evidence[30 31] was inspected and concepts are drawn on to form the resultant programme theory in figure 3 which shows two main channels: patient copies of letters and patient personalised letters. Contexts for when patients receive letters still contained five key contexts for when this intervention does work but context details were modified. Previously, the theory had four key contexts for when the intervention is theorised not to work; these were updated to include the new context of judgemental language in relation to social behaviour (CMOC53). Outcomes of patients receiving separate personalised letters were modified; new negative outcomes were overly 'basic' content and perceived potential secrecy between clinicians if they are sending and receiving separate letters. 'Patient choice' was still a key influencer for likelihood of beneficial outcomes, and contextual influences such as resource provision and directives (CMOC49) were determiners of patients being given a choice of letter receipt (CMOC52).

## DISCUSSION
### Summary of findings

We undertook a realist evaluation[19 28 55 56] to explore patient, GP, and hospital clinician experiences of written discharge communications and hence test and refine the programme theory from our previous realist review.[14] The modified programme theory (figure 3) maps out how patients receiving discharge letters works in specific contexts leading to different positive and negative outcomes. Positive outcomes and positive pathway components are indicated in figure 3 via green coloured text boxes whereas negative outcomes and negative pathway components are indicated in red. Any neutral components or those which can be either positive or negative (eg, attitudes of clinicians) are in black. Analysis of the multi-perspective discharge events led to the emergence of findings not found in our previous review. Several changes to the initial theory were made to include 10 CMOC modifications and the addition of 7 new CMOCs not found through previous literature searching. No CMOCs were removed. Key contexts for positive outcomes included: letters written in plain English, lay explanations for jargon, written and verbal information provided, no new information in letter and patient given choice of letter receipt.

While benefits[42 57] and drawbacks[54 58] of patients receiving discharge letters have been previously suggested, our study adds an understanding of *how* patients receiving letters *works* through outlining the important contexts and associated mechanisms that

explain outcome patterns.[59 60] In addition, the multi-perspective analysis provided possible explanations for previously reported discrepancies identified through our realist review.[14] One example of a discrepancy was that past work highlighted conspicuously inconsistent rates of patient understanding.[12 41 47 48 61 62] Data from this study revealed that even in cases where clinicians expressed concerns, patients generally reported to have understood the letter and found it useful. Furthermore, patients often preferred receiving the same letter as the GP rather than a separate letter. Another disparity was in relation to 'negative outcomes'. A common clinician concern within the study and past literature[13 38 44] was that patients receiving letters may cause anxiety and harm. However, literature also reported that patients may find letters useful.[12 45 48] Our method highlighted that in several cases where clinicians had concerns, patients who received letters tended to emphasise the positive effects (eg, increased knowledge). Indeed, patients stressed negative outcomes in contexts where they *had not* rather than *had* received letters. Some patients reported that receiving the letter alleviated anxiety thereby supporting their well-being through informing them of their admission, and any next steps, as well as providing reassurance that their GP was updated.

### Strengths and weaknesses of the study

We followed Realist And Meta-narrative Evidence Syntheses: Evolving Standards (RAMESES) for realist evaluation[29 63] and completed the Consolidated criteria for Reporting Qualitative research checklist by Tong *et al*.[64] To the best of our knowledge, this is the first realist study to triangulate matched perspectives of patients and clinicians in relation to specific discharge letters. This allowed reconciliation of disparities in the literature and so enabled refinement of the programme theory. Grounding the research in realist theory strengthened the applicability of findings as it facilitated an understanding of not just whether patients should receive letters, but how this practice may 'work', as well as in what contexts and why.[16 17]

As with other realist evaluations,[65] the results and findings are intended to have wide applicability to other settings, in this case, settings where adults may receive hospital discharge letters. However, it is important to note the contexts and those groups who were excluded or were under-represented in this study. The exclusion criteria restricted the programme theory such that evidence relating to children, solely to mental health, and those lacking capacity to consent was not considered. Moreover, participation bias may have resulted in the views of ethnic minorities and other marginalised groups being under-represented. The main weakness of the study was the small sample sizes in terms of numbers of patients, sociodemographic diversity of the patients and range of conditions; for many of the discharge letters it was not possible to form a complete quartet. The study fell short of the target of building 30 quartets; the primary reason

for this was under-recruitment of HPs. The low response rate of HPs was likely impacted by their lack of available time, our survey recruitment strategy, hospital rotations, and the time lapse between the practitioner writing the letter and receiving the survey invitation. The programme theory would have benefitted from being informed by a larger and more diverse sample of primary evidence. The matched cases relate to a specific geographic area and hence will not have reflected the full range of hospital discharge communication practices that are present nationally. Analysis cannot be considered to be wholly objective due to the influence of researcher identity.[66] Therefore, 'reflexivity' was practised throughout the research to reduce but not eradicate bias.[66 67] Reflexivity was practised through keeping a research diary and regular research team discussion and reflection. Data analysis was also limited by the available evidence which was thin in relation to: dictating letters, the cost of patients receiving letters, doctor–patient relationships, and reasons for variation of practice. Further research is needed to explore these areas as well as the relevance of the programme theory to excluded and under-represented groups, such as those without capacity and children.

### Meaning of the study: implications for clinicians and policy-makers

The programme theory generated by this study draws on our previous review and primary data, and hence reflects evidence from 16 countries and over 16 000 participants. As such, the theory has both national and international relevance and is likely to be applicable to different healthcare settings. It generally supports policies[7–9 11] that patients should be offered copies of letters between physicians. Although sending patients' letters, to include discharge letters, has been recommended practice for almost 20 years,[7] uptake remains inconsistent.[12–14] Although national guidelines exist,[7–10 68 69] each hospital may have its own discharge policy; this means that patients may have different discharge experiences and receive different discharge communications depending on the hospital, discharging physician and reason for admission, as exemplified in this study. This needs to be addressed with more standardised practices which account for individual preferences and are grounded by *patient choice* with the exception of where there is a risk of 'harm', as defined in guideline documents.[7] Patients have a right to receive their letters[11] and should not be denied the opportunity to receive letters based on the perception that their understanding may be low. Although patients may have limited health literacy, they demonstrated resourcefulness and resilience for accessing letter content by looking up unknown terms on the internet and also appreciated that understanding the important features and main directives of a letter do not necessarily involve comprehending every word. Thus, patient understanding is perhaps greater than perceived and the presence of clinical terminology alone is not reason enough to exclude patients from communications. Overall, our study found

that negative outcomes more commonly manifested when patients had not received letters, rather than when they had. This included contexts where the clinicians had concern about patient understanding and yet the patient reported to have found the letter of value. It may be inferred that within certain contexts, clinician concerns about patients receiving letters are perhaps unfounded. Thus, clinician attitudes and risk-averse behaviour may be acting as a barrier to uptake of this practice.

This research has provided a modified programme theory which demonstrates how policy makers and clinicians may effectively involve patients in their care through provision of written communications. Our theory outlines how both positive and negative outcomes may be produced through this intervention and highlights the importance of contextual considerations.[56] As outlined in previous realist evaluations,[60] an advantage of this approach is the relevance of the resultant theory to policy makers as it informs how policy may be adapted to particular purposes and the specific contexts needed to achieve the desired outcomes. An example is the importance of the contextual factor 'patient choice of letter receipt' to producing positive outcomes; this is of relevance to policy makers as it explains how best practice of patients receiving letters may be adapted to 'work' and how research may be implemented into practice and policy. Nonetheless, future work should endeavour to test and refine the programme theory through interrogation of new evidence and measurement of primary and secondary outcomes. This will support the development of interventions that lead to more effective communication between hospital and primary care health professionals, and hence positive patient outcomes.

## CONCLUSION

Sharing information and effective discharge communication with patients should be a priority to improve patient experience and the safety of patient care. This study has yielded insights into ways in which practices of patients receiving discharge letters could be improved to enhance patient experience and outcomes. Key findings were: clinicians may underestimate patients' capacity to comprehend discharge letters, patient choice is important for positive outcomes, absence rather than the presence of information may be more associated with negative outcomes, and that clinician attitudes may be acting as a barrier to patients receiving letters. Our programme theory draws on previous research and experiences of clinicians and patients. The theory has potential for use in different healthcare contexts and as a framework for policy development on patient discharge.

**Acknowledgements** The authors would like to express thanks to the GPs, patients, and hospital practitioners who informed the study design as well as those who took part in the study. We would like to thank the West Midlands Clinical Research Network for their support with research recruitment. We would also like to thank the Institute of Advanced Study at the University of Warwick for their support during the write-up of this paper. Finally, we would like to thank Hugh Rayner for providing feedback on an earlier version of the manuscript.

**Contributors** KW, ES, SS and JD substantially contributed to the design and conception of the work and collaborated with KW on data interpretation. KW drafted the manuscript which was then critically revised by JD, ES and SS. All authors approved the final manuscript for publication and agree to be accountable for all aspects of the work.

**Funding** This work was supported by the Economic and Social Research Council (grant number ES/J500203/1) and Clinical Commissioning Groups of Coventry & Rugby and South Warwickshire. Funding for the open access charges for publication of this manuscript was provided by the UKRI fund. The funders had no involvement in analysis or preparation of the manuscript.

**Competing interests** None declared.

**Patient consent for publication** Not required.

**Ethics approval** Ethics approval was granted by the UK Health Research Authority in July 2017 (Integrated Research Application System ID: 219871, Research Ethics Committee reference: 17/WM/0170, sponsor: University of Warwick).

**Provenance and peer review** Not commissioned; externally peer reviewed.

**Data availability statement** All data which can be made available are included in the article or uploaded as supplementary information. The narrative summaries for data are provided as supplementary files. The full raw datasets for the study are not publicly available due to the sensitive and identifiable nature of the data. Despite names and other identifiers being removed, the in-depth nature of case information may mean that participants could be identified.

**ORCID iDs**
Katharine Weetman http://orcid.org/0000-0003-0377-3933
Jeremy Dale http://orcid.org/0000-0001-9256-3553

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
