## [Reviewer comments · BMJ Open]

ARTICLE DETAILS

TITLE (PROVISIONAL)	The Discharge Communication Study: A Realist Evaluation of Discharge Communication Experiences of Patients, General Practitioners, and Hospital Practitioners, Alongside a Corresponding Discharge Letter Sample
AUTHORS	Weetman, Katharine; Dale, Jeremy; Scott, Emma; Schnurr, Stephanie

VERSION 1 – REVIEW

REVIEWER	Scott, Jason Northumbria University, Faculty of health and life sciences
REVIEW RETURNED	07-Dec-2020

GENERAL COMMENTS	This was a very interesting piece of research that has clearly been conducted to a high standard. I do have a number of comments and suggestions (mostly in relation to how the methods are reported) that require addressing before I can recommend acceptance: Title and Abstract GP shouldn't be abbreviated. Data is the plural or datum, so it should be 'data were'. Introduction The research is excellently situated within both current evidence and policy, and has been written for an international audience. There is a minor typo: (background, paragraph 1). "Reasons for this inconsistently..." should be "inconsistency". Methods You are reporting one aspect of a larger study here, which I have no problems with and can understand why you are reporting the data like this. I am however somewhat unclear about how it is distinguished from the other 'studies'. The COREQ checklist that has been completed almost entirely refers to box 1 (page 4), which is reporting data from the other published studies. As this is being reported as an individual study, the manuscript should contain sufficient reporting detail to be standalone. The methods section at present doesn't adequately describe the methods utilised in relation to this specific manuscript. There is a single section for 'recruitment and data collection', but it's not clear which data are being reported on here, rather than what was collected in other parts of the study. If you are re-interpreting data collected across all of the the other reported studies, then this needs to be made more explicit, with a summary of the data provided within the text and under appropriate headings (the box should supplement this, not replace it). I'm also unclear about what the data utilised in this manuscript
---

	actually consisted of. For instance, in the methods it's reported that GPs graded discharge letters as successful or unsuccessful; this is an important consideration for this paper, but being buried away in another manuscript makes it difficult to follow. I understand the difficulties in reporting data across multiple studies and some assumptions can fairly be made, such as readers having a general understanding of semi-structured interviews / surveys without reading the full papers, but this can't be made for something as nuanced as GP ratings. I did think that the analysis process was very well explained with strong justifications throughout. Results The findings are generally very well presented, and I particularly like how you have supported your findings with links through to both the CMO model and the developed narratives. I have no comments on the findings other than table 2 and figure 3 (the latter is the only reason I ticked 'no' to question 10 on whether results are presented clearly): In table 2, please report the specialties if able to do so. Also, is 'role' the correct word in the final row? A more accurate description may be 'grade'. Someone working at a consultant grade may have several different roles. Figure 3 was difficult to read due to the size and quality of the text. This may have been the journal's conversion process, so please pay particular attention to this at proofing stage. Notably within the 'contexts' column in the centre of the diagram it looked like boxes were overlapping and possibly(?) cutting some text off. I was also a little unsure why the contexts had two separate 'yes' answers between contexts and mechanisms. Perhaps some thematic grouping is required to distinguish them for the reader? Finally, are the different arrow sizes representative of the strength of link? If not, you should standardise these throughout as it looks messy at present. Discussion The discussion, as with the introduction, is written to a very high standard. The only thing I recommend the authors to include (likely in the final paragraph before the conclusion) is a statement about how future research needs to test the proposed programme theory through measurement of the primary and secondary outcomes. As part of this it would be of interest to readers for the authors to reflect across the whole project whether any of the outcomes would take priority if such measurement were to occur – did you get a feel for this from participants?
--	---

REVIEWER	Le Meur, Nolwenn Ecole des Hautes Etudes en Sante Publique
REVIEW RETURNED	08-Dec-2020

GENERAL COMMENTS	The proposed manuscript entitled "The Discharge Communication Study: a realist evaluation of discharge communication experiences of patients, GPs, and hospital practitioners, alongside a corresponding discharge letter sample" seems to be the latest of a series of papers on the research question by the same authors. The paper is well documented with important and up to date bibliography. The research question and results are interesting for those in the field. The paper proposed practical recommendations to
---

	health services. Although my minor modifications might improve the paper and its reading.  - In table 2 what would be the other types of admission, apart from inpatient and outpatient. I could be added at the bottom of the table. - Figure 2 is not clear. Does it mean that the recruitment was not specific to this study and that it is the results of study 2 to 4? - In Figure 3 colours would be appreciated to distinguish between positive and negative components of the programme theory. In general the captions of the figures are succinct and not helping the reader. The figures are not really self-content. Finally, I understand that the study was based on qualitative interviews and thus a small sample but could you assess the questions below?:  - Was there a difference in perception or outcomes between gender of patients or health care professionals ? - Was there a difference in perception or outcomes between age groups of patients or health care professionals ? - Was there a difference in perception or outcomes between diseases ?
--	---

VERSION 1 – AUTHOR RESPONSE

Reviewer: 1

Dr. Jason Scott, Northumbria University

Comments to the Author:

Dear Dr Weetman and colleagues,

This was a very interesting piece of research that has clearly been conducted to a high standard.

Thank you for your comment and positive feedback.

I do have a number of comments and suggestions (mostly in relation to how the methods are reported) that require addressing before I can recommend acceptance:

Title and Abstract

GP shouldn't be abbreviated. Data is the plural or datum, so it should be 'data were'.

Thank you, these have been corrected.

Introduction

The research is excellently situated within both current evidence and policy, and has been written for an international audience. There is a minor typo: (background, paragraph 1). "Reasons for this inconsistently..." should be "inconsistency".

Thank you, this has been corrected.

Methods

You are reporting one aspect of a larger study here, which I have no problems with and can understand why you are reporting the data like this. I am however somewhat unclear about how it is distinguished from the other 'studies'. The COREQ checklist that has been completed almost entirely refers to box 1 (page 4), which is reporting data from the other published studies. As this is being reported as an individual study, the manuscript should contain sufficient reporting detail to be standalone. The methods section at present doesn't adequately describe the methods utilised in

relation to this specific manuscript. There is a single section for 'recruitment and data collection', but it's not clear which data are being reported on here, rather than what was collected in other parts of the study. If you are re-interpreting data collected across all of the other reported studies, then this needs to be made more explicit, with a summary of the data provided within the text and under appropriate headings (the box should supplement this, not replace it).

For this study, we undertook a secondary analysis of a sub-set of the data that was collected across all of the other reported studies. To make this clearer we have added the following two sections of text to the methods section on page 6:

"For this study, we re-interpreted data collected across all of the other studies."

"The target was to build 30 quartet cases through recruiting at least 30 GPs, patients and hospital practitioners (HPs) (target n=90). Trio and quartet participants were not separately recruited from other studies within the project; instead, cases were built through the participant recruitment and data collection across all studies for the discharge communication project (see figure 2). Once participant data across studies were matched into trio and quartet cases, findings and data were subjected to a secondary level data analysis using a realist approach described below. This allowed highlighting of data convergence and divergence as well as the emergence of new findings which only become apparent through juxtaposition."

In response to this comment, the methods section has been restructured and substantially revised from pages 5-6. The COREQ checklist has been updated to reflect this. New headings have also been added to the METHODS section to reflect those used in the study protocol. The new Methods section reads as follows:

"Study design

This study was a secondary analysis of a subset of data from the Discharge Communication Study, an exploratory mixed methods study based in the West Midlands, United Kingdom (UK) (21); box 1 gives a brief summary of papers linked to the Discharge Communication Study. The intervention under scrutiny 'patients receiving discharge letters' was defined by the team as 'the patient being given or sent any form of written (paper or digital) hospital discharge communication; this could be a direct copy, patient-directed letter, or a combination.' Broadly, the data comprised three elements: (1) GP sampling and rating of discharge letters ("successful" or "unsuccessful") and narrative interviews, (2) semi-structured interviews with patients to whom the letters related, (3) survey of hospital practitioners who wrote the sampled letters.

Settings

The setting for the study is outlined in the published study protocol (21). It involved four hospital trusts and a diverse range of 18 GP practices in the West Midlands.

-Recruitment and data collection

Recruitment and data collection took place, as detailed in previous publications (21-23) between August 2017 and September 2018. In brief, GPs were asked to screen (see table 1 for screening criteria) and select a sample of recently received discharge letters according to what they considered to be "successful" or "unsuccessful" letter exemplars; for each letter, GPs were asked to complete the selection proforma (supplementary file 1) and rate the letters "successful" or "unsuccessful". There were no set criteria for letter ratings as the selection was based on each participating GP's interpretation of what makes a successful or unsuccessful discharge letter. This purposive (24-26) letter sampling approach was intended to increase sample diversity and address the research questions within dichotomous contexts. All GPs involved in letter sampling were then invited to take part in a "narrative" (27, 28) interview or focus group with KW (see supplementary file 2 for interview

guide). All patients associated with the sampled discharge letters were sent an invitation pack by their GP practice; this invited them to take part in an audio recorded semi-structured interview with KW (see supplementary file 3 for interview guide). Finally, the hospital professionals who wrote or signed the sampled discharge letters were sent an invitation pack by the research team; this invited them to take part in a survey on their evaluation of the discharge letter they wrote, their current practices, and their views about how discharge communication processes may be improved. Packs were sent by post and email as well as being internally distributed by hospital sites.

For this study, we re-interpreted data collected across all of the other studies. This involved a secondary analysis of a subset of the data which was drawn from sampled discharge letters that could be “matched” to at least two other dataset perspectives. Study specific ID codes allocated to the letters allowed cross-matching with participants to build multiple viewpoint cases termed “quartets” (mapping together four elements if complete, or “trios” if only one perspective missing - see figure 1). The target was to build 30 quartet cases through recruiting at least 30 GPs, patients and hospital practitioners (HPs) (target n=90). Trio and quartet participants were not separately recruited from other studies within the project; instead, cases were built through the participant recruitment and data collection across all studies for the discharge communication project (see figure 2). Once participant data across studies were matched into trio and quartet cases, findings and data were subjected to a secondary level data analysis using a realist approach described below. This allowed highlighting of data convergence and divergence as well as the emergence of new findings which only became apparent through juxtaposition.”

The text describing figure 2 has been modified as follows (results section, page 8):

“Figure 2 shows how data collection across all studies for the discharge communication project led to the formation of 26 trio cases (1 GP and HP, 3 patient and HP, 22 patient and GP) and 10 quartet cases (patient, GP, and HP).”

In addition, the legend for figure 2 has been amended for increased clarity and is now:

“Figure 2 Recruitment uptake across studies for the project to show how trio and quartet cases were formed”

I’m also unclear about what the data utilised in this manuscript actually consisted of. For instance, in the methods it’s reported that GPs graded discharge letters as successful or unsuccessful; this is an important consideration for this paper, but being buried away in another manuscript makes it difficult to follow. I understand the difficulties in reporting data across multiple studies and some assumptions can fairly be made, such as readers having a general understanding of semi-structured interviews / surveys without reading the full papers, but this can’t be made for something as nuanced as GP ratings.

We agree and so the following further sentences have been added within the METHODS section (page 5):

“GPs were asked to screen (see table 1 for screening criteria) and select a sample of recently received discharge letters according to what they considered to be “successful” or “unsuccessful” letter exemplars; for each letter, GPs were asked to complete the selection proforma (supplementary file 1) and rate the letters “successful” or “unsuccessful”. There were no set criteria for letter ratings as the selection was based on each participating GP’s interpretation of what makes a successful or unsuccessful discharge letter. This purposive (24-26) letter sampling approach was intended to increase sample diversity and address the research questions within dichotomous contexts.”

I did think that the analysis process was very well explained with strong justifications throughout. Thank you for your comment and positive feedback.

Results

The findings are generally very well presented, and I particularly like how you have supported your

findings with links through to both the CMO model and the developed narratives. I have no comments on the findings other than table 2 and figure 3 (the latter is the only reason I ticked 'no' to question 10 on whether results are presented clearly): In table 2, please report the specialties if able to do so.

The specialties are now reported in table 2.

Also, is 'role' the correct word in the final row? A more accurate description may be 'grade'. Someone working at a consultant grade may have several different roles.

We agree and so this has been changed to "grade" throughout.

Figure 3 was difficult to read due to the size and quality of the text. This may have been the journal's conversion process, so please pay particular attention to this at proofing stage. Notably within the 'contexts' column in the centre of the diagram it looked like boxes were overlapping and possibly(?) cutting some text off.

Thank you for highlighting this issue. The figure text is fully visible on our uploaded version of the figure. We will check this at proofing stage to ensure no text is cut.

I was also a little unsure why the contexts had two separate 'yes' answers between contexts and mechanisms. Perhaps some thematic grouping is required to distinguish them for the reader? Finally, are the different arrow sizes representative of the strength of link? If not, you should standardise these throughout as it looks messy at present.

The "yes" elements of the contexts have now been removed for clarity. Thematic grouping has not been used as a realist approach to devising the programme theory has been followed. Furthermore, we believe the figure may become confounded by adding further details and groupings. Arrow sizes are not representative of strength and so in response to your comment these have been standardised throughout.

Discussion

The discussion, as with the introduction, is written to a very high standard.

Thank you for your positive feedback.

The only thing I recommend the authors to include (likely in the final paragraph before the conclusion) is a statement about how future research needs to test the proposed programme theory through measurement of the primary and secondary outcomes. As part of this it would be of interest to readers for the authors to reflect across the whole project whether any of the outcomes would take priority if such measurement were to occur – did you get a feel for this from participants?

A statement to this effect has been added above the CONCLUSION on page 16. We agree that prioritization of outcomes would be of interest to readers. We do not feel, based on our data, that specific outcomes should be selected as a priority. Thus, the new section reads as follows: "Nonetheless, future work should endeavour to test and refine the programme theory through interrogation of new evidence and measurement of primary and secondary outcomes. This will support the development of interventions that lead to more effective communication between hospital and primary care health professionals, and hence positive patient outcomes."

Reviewer: 2

Dr. Nolwenn Le Meur, Ecole des Hautes Etudes en Sante Publique Comments to the Author:

The proposed manuscript entitled "The Discharge Communication Study: a realist evaluation of discharge communication experiences of patients, GPs, and hospital practitioners, alongside a corresponding discharge letter sample" seems to be the latest of a series of papers on the research

question by the same authors. The paper is well documented with important and up to date bibliography. The research question and results are interesting for those in the field. The paper proposed practical recommendations to health services. Although my minor modifications might improve the paper and its reading.

Thank you for your comments and positive feedback.

- In table 2 what would be the other types of admission, apart from inpatient and outpatient. I could be added at the bottom of the table.

In response to this comment the following has been added at the bottom of the table:

“*other may include but not be limited to admission types such as accident and emergency visit, day case procedure, or speciality assessment unit visit.”

- Figure 2 is not clear. Does it mean that the recruitment was not specific to this study and that it is the results of study 2 to 4?

Recruitment for studies 2-4 built the trio and quartet cases. The cases have been subjected to a secondary level analysis but are not distinct from the participants in studies 2-4. Matching the cases and analysing them comparatively through realist evaluation is the primary contribution of this study. The methods section has been revised to reflect this. In addition, to make this clearer, we have re-labelled figure 2 as follows:

“Figure 2 Recruitment uptake across studies for the project to show how trio and quartet cases were formed”

- In Figure 3 colours would be appreciated to distinguish between positive and negative components of the programme theory.

In response to this comment, colours have been added to the revised programme theory. The colours reinforce the messages in the theory and therefore the figure would still be accessible and comprehensible if printed in black and white and/or to persons with colour blindness or visual impairments. To clarify the colour meanings for readers, the following text has also been added to the main manuscript in the DISCUSSION on page 13:

“Positive outcomes and positive pathway components are indicated in figure 3 via green coloured text boxes whereas negative outcomes and negative pathway components are indicated in red. Any neutral components or those which can be either positive or negative (e.g., attitudes of clinicians) are in black.”

In general, the captions of the figures are succinct and not helping the reader. The figures are not really self-content.

Thank you for this comment. We agree and so figure legends have been revised as follows:

Figure 1 Multiple-perspective “quartet” case wherein comparisons occur between experiences associated with the same discharge letter

Figure 2 Recruitment uptake across studies for the project to show how trio and quartet cases were formed

Figure 3 Resultant programme theory that maps out how patients receiving discharge letters works (or not)

Finally, I understand that the study was based on qualitative interviews and thus a small sample but could you assess the questions below?:

- Was there a difference in perception or outcomes between gender of patients or health care professionals ?
- Was there a difference in perception or outcomes between age groups of patients or health care professionals?
- Was there a difference in perception or outcomes between diseases?

We agree these assessments are of interest. However, we looked at these components and due to the small sample size and the number of letters/participants within each group (e.g. only one Cardiology quartet), no patterns or differences were identified qualitatively and quantitative statistical analysis on a sample of this nature, particularly due to the number of uncontrolled variables, would have been inappropriate. We did find statistical differences between the content and quality of the discharge letters and outcomes but this is the topic of our final paper linked to the study (which has recently been drafted and is currently under review).

VERSION 2 – REVIEW

REVIEWER	Scott, Jason Northumbria University, Faculty of health and life sciences
REVIEW RETURNED	01-Feb-2021

GENERAL COMMENTS	Thank you for addressing and/or responding to the previous comments. I have no further comments to make and will keep an eye out for this being published and your other paper.
---

REVIEWER	Le Meur, Nolwenn Ecole des Hautes Etudes en Sante Publique
REVIEW RETURNED	10-Feb-2021

GENERAL COMMENTS	Thank you for considering my comments and improving the clarity of the papers. I only have a few comments left on the form: Why in the title "General Practitioners", and elsewhere in different paragraphs, are written with capital letters? And why not hospital practitioners? I understand that general practitioners can be abbreviated GP but I am not sure that in full letters capital letters should be used. Figure3: I think that the colouring of the boxes should be explained in the caption with a similar paragraph as the one added in the discussion ("Positive outcomes and positive pathway components are indicated in green coloured text boxes whereas negative outcomes and negative pathway components are indicated in red. Any neutral components or those which can be either positive or negative (e.g., attitudes of clinicians) are in black.) Figure 3: As background of the figure is black on the pdf, fine black arrows between boxes are no more visible. For example, arrows are no more visible between "Patient does not receive copy of discharge letter" and the 2 boxes below (Reduced patient autonomy and knowledge - Patient satisfied if choice acknowledged)
--

VERSION 2 – AUTHOR RESPONSE

Reviewer: 1

Dr. Jason Scott, Northumbria University

Comments to the Author:

Dear authors,

Thank you for addressing and/or responding to the previous comments. I have no further comments to make and will keep an eye out for this being published and your other paper.

Thank you for your comments and this positive feedback on the revised version.

Reviewer: 2

Dr. Nolwenn Le Meur, Ecole des Hautes Etudes en Sante Publique

Comments to the Author:

Dear authors,

Thank you for considering my comments and improving the clarity of the papers. I only have a few comments left on the form:

Why in the title "General Practitioners", and elsewhere in different paragraphs, are written with capital letters? And why not hospital practitioners? I understand that general practitioners can be abbreviated GP but I am not sure that in full letters capital letters should be used.

The capitalisation in the title for both types of practitioners has been retained but in response to this comment has been removed in the main body of the manuscript so that both general practitioner and hospital practitioner are consistently lowercase throughout.

Figure3: I think that the colouring of the boxes should be explained in the caption with a similar paragraph as the one added in the discussion ("Positive outcomes and positive pathway components are indicated in green coloured text boxes whereas negative outcomes and negative pathway components are indicated in red. Any neutral components or those which can be either positive or negative (e.g., attitudes of clinicians) are in black.)

This caption has been added to figure 3 as requested.

Figure 3: As background of the figure is black on the pdf, fine black arrows between boxes are no more visible. For example, arrows are no more visible between "Patient does not receive copy of discharge letter" and the 2 boxes below (Reduced patient autonomy and knowledge - Patient satisfied if choice acknowledged)

To address the comment, the file has been resaved as a PDF rather than TIFF which has removed the black background and replaced with white.